# Evaluation of the Protection Effectiveness of Natural Protected Areas on the Qinghai–Tibet Plateau Based on Ecosystem Services

**DOI:** 10.3390/ijerph20032605

**Published:** 2023-01-31

**Authors:** Mengdi Fu, Jun Wang, Yanpeng Zhu, Yuanyuan Zhang

**Affiliations:** 1State Key Laboratory of Environmental Criteria and Risk Assessment, Chinese Research Academy of Environmental Sciences, Beijing 100012, China; 2Center for Biodiversity and Nature Reserve, Chinese Academy of Environmental Planning, Beijing 100043, China; 3Beijing Milu Ecological Research Center, Beijing Biodiversity Conservation Research Center, Beijing 100076, China

**Keywords:** natural protected area, ecosystem service, protection effectiveness, geodetector, Qinghai–Tibet Plateau

## Abstract

Evaluating the protection effectiveness of natural protected areas is an important step in successful management. Adopting 330 natural protected areas on the Qinghai–Tibet Plateau as research subjects, the regional dominant ecosystem service function was selected, and various temporal and spatial analysis methods were employed to analyze the evolution characteristics and influencing factors of ecosystem service patterns from 2000 to 2020. Our results indicated that (1) the water conservation function stabilized after fluctuation and decline, the soil conservation function fluctuated upward, and the windbreak and sand fixation function exhibited an increase after a decreasing fluctuation. (2) The protection effectiveness of25 protected areas significantly improved, that of 151 protected areas improved, that of 84 protected areas stabilized, that of 56 protected areas worsened, and that of 14 protected areas significantly worsened. (3) The top three influencing factors in descending order were precipitation change > altitude > mining area density. (4) Remarkable protection results were achieved in national protected areas, established management institutions, earlier established areas (before 2000), and areas exhibiting alow built-up area density (<0.75%) and low mining density (<1%). Our study provides technical support for the construction and management of protected areas and improvement in ecosystem service functions on the Qinghai–Tibet Plateau.

## 1. Introduction

Globally, natural protected areas are recognized as the most effective means of natural protection, protecting the world’s important natural ecosystems and biological resources. The evaluation of their effectiveness is the focus of muchcurrent research [1]. In recent years, different methods have been used to carry out protection effectiveness evaluation studiesfor different regions and different types of natural protected areas, and the evaluation results have beenconsidered to inform management decisions. For example, Timko and Satterfield [2] assessedthe effectiveness of national parks and naturereserves in Canada, Australia, and South Africa in terms of protecting biodiversity and ecosystem processes. Zheng et al. [3] evaluated the protection effectiveness of 91 national wetland nature reserves in China and found that 79% (area) of the nature reserves exhibited poor protection effectiveness. Joppa and Pfaff [4] evaluatedthe protection effectiveness of natural protected areas in 147 countries. Xin et al. [5,6] developed an evaluation indicator system for the protection effectiveness of China’s desert and grassland nature reserves and establisheddemonstration applications. Yang et al. [7] constructed an evaluation indicator system for the protection effectiveness of national nature reserves for migratory birds and conducted trial evaluations in the Qinghai Lake and East Dongting Lake national nature reserves. Khan et al. [8] evaluated the effectiveness of Meghalaya State’s protected area network in terms of biodiversity protection based on plant diversity indicators. Tissot et al. [9] evaluatedthe effectiveness of marine protected area networks in western Hawaii based on fish population and benthic habitat data. Sobhani et al. [10] assessedthe spatial and temporal changes in natural capital in a typical semiarid protected area based on an ecological footprint model, to evaluate the sustainability of the use of natural resources.Rahmadyani et al. [11] assessed stakeholders’perceptions of the value of coral reef ecosystem services in the GiliMatra Marine Tourism Park. Whilemost researchers have focused on the protection effectiveness evaluation of nature reserves [12,13], veryfew have focused on the protection effectiveness evaluation of other protected areas, such as national parks, forest parks, wetland parks, and geoparks. Considering the availability of data, most research focuses on national and provincial natural protected areas, and thus far, there is insufficient research on the protection effectiveness of county-level natural protected areas. To maintain the integrity of natural ecosystems, there is a lack of studies that evaluateprotection effectiveness by targetingnatural protected area networks with geographic units as the main body. In addition, protection effectiveness evaluation is mostly based on qualitative evaluation, which is insufficiently connected with the management of natural protected areas, and the evaluation results are difficult to apply to the management process.

The Qinghai–Tibet Plateau is located in the middle of the Asian continent. It is an important ecological security barrier, a germplasm bank of alpine organisms, and a strategic resource reserve base in China [14]. This areaoccupies an irreplaceable position in the process of maintaining national ecological security and sustainable development of the Chinese nation. By the end of 2020, 330 natural protected areas of various types and at all levels were established on the Qinghai–Tibet Plateau, covering an area of 88.34 million hm^2^ (excluding overlapping parts) and accounting for 34.20% of the total plateau area, representing the unique and fragile ecosystems and rare species resources of the plateau. These protected areas play an important role in protecting biodiversity and enhancing ecosystem services. An objective evaluation of the extent to which natural protected areas have achieved protection goals is important for improving the management efficiency of natural protected areas. In this study, the dominant ecosystem service function on the Qinghai–Tibet Plateau was selected, an evaluation of the protection effectiveness of natural protected areas was conducted based on ecosystem services, and the influencing factorswere identified to inform the development and management of Qinghai–Tibet Plateau natural protected areas, enhance regional ecosystem services, and safeguard national ecological security.

Ecosystem service assessment is fundamental for ecosystem management and decision-making processes. At the early stage of the research on ecosystem services, forest ecosystems [15] were mainly adopted as the evaluation object, followed by research on grasslands, deserts, wetlands, oceans, and farmland ecosystems. In recent years, scholars have performed a large number of studies involving ecosystem service assessment at different scales, such as countries [16,17], regions [18,19], basins [20,21], and administrative units [22], to provide an important basis for ecological asset management and ecological compensation policy formulation. Other scholars have carried out related research on ecosystem services in national key ecological function areas. For example, Liu et al. [23] quantitatively analyzed the temporal and spatial distribution pattern and change characteristics of the ecosystem service value in 25 national key ecological function areas after the implementation of transfer payments. The results showed that remarkableecological protection and engineering construction results had been achieved. Zhou et al. [24] evaluated the temporal and spatial changes in ecosystem services in the water source area of the middle route of the South–North Water Transfer Project and established ecological compensation standards and apportionment mechanisms. With the goal of improving regional ecosystem services, scholars have proposed a framework for ecological protection and restoration of forestland, wetland, grassland, and cropland areas based on ecosystem services to achieve overall protection, quality improvement, and pattern optimization of river basin ecosystems [25].

At present, a global system of protected areas with relatively complete types and relatively complete functional layouts has been established [26]. However, there remain problems such as the outstanding contradiction between the protection and development of protected areas, the imperfect ecological compensation system, and the inability to evaluate the benefits of ecological projects. The ecosystem service assessment of protected areas provides an effective way to solve the above problems. In this study, the dominant ecosystem services on the Qinghai–Tibet Plateau were selected to quantitatively evaluate the degree of change, spatial pattern evolution, and agglomeration characteristics of ecosystem services at all levels and types of natural protected areas from 2000 to 2020 using the Geodetector model, thereby analyzing the influencing factors and providing a basis for protection countermeasure improvement.

## 2. Materials and Methods

### 2.1. Study Area

The Qinghai–Tibet Plateau comprises towering mountains, vast plateaus, scattered lakes, and numerous water systems. The terrain slopes from northwest to southeast, with an average elevation of more than 4000 m. The annual average temperature is 1.37°C, and the total solar radiation ranges from 5400 to 8000 MJ/(m^2^∙a). The annual precipitation variesbetween 20 and 4500 mm, with an extremely uneven spatial distribution. The Qinghai–Tibet Plateau is the birthplace of the Yellow, Yangtze, Lancang–Mekong, Brahmaputra, Nu-Salween, Dulong, Tarim, Ganges, Indus, and Amu Darya Rivers. It occupies a strategic position in terms of the generation, storage, and migration of water resources in China and Asia. There are 36,000 contemporary glaciers, and they comprise the second-highestconcentration of glaciers worldwide after polar ice caps. The soil is dominated by alpine meadow soil, alpine grassland soil, alpine cold desert soil, and subalpine meadow soil, with the characteristics of a thin soil layer, simple layer features, strong coarse bone, low degree of weathering, and lowcorrosion resistance. The types of ecosystems are complex and diverse, with primary forests and natural secondary forests accounting for more than 96% of the total forest area.

The Qinghai–Tibet Plateau involves six Chinese provinces: Tibet Autonomous Region, Xinjiang Uygur Autonomous Region, Qinghai Province, Gansu Province, Sichuan Province, and Yunnan Province (Figure 1). There are five main types of natural protected areas, including nature reserves, forest parks, wetland parks, geoparks, and desert parks. Among them, there are 174 nature reserves. The total human population in these natural protected areas is 1.41 million people, and the population density is 1.6 people/km^2^.

### 2.2. Data Sources

The data on natural protected areasin China were provided by the National Forestry and Grassland Data Center. Meteorological data, including temperature, relative humidity, precipitation, evaporation, wind speed, and sunshine hours, were supplied by meteorological stations in the study area and surrounding regions. The meteorological data were spatially interpolated using ANUSPLIN [27]. The potentialevapotranspiration (ET0) was calculated using the revised version of the Penman–Monteith model ofthe United Nations Food and Agriculture Organization (FAO) [28,29,30]. The adopted snow depth data originated from the long-term snow depth dataset of China [31], which is provided by the Environmental and Ecological Science Data Center for West China. Land-use data were retrieved from China’s Multi-Period Land Use Land Cover Remote Sensing Monitoring Dataset (CNLUCC) [32] with a spatial resolution of 30 m. Soil attribute data were obtained from the Chinese soil type spatial distribution dataset. Vegetation coverage data were acquired from the Moderate-Resolution Imaging Spectroradiometer (MODIS) vegetation index (MOD13Q1). The maximum value composite (MVC) method [33] was used to synthesize the normalized vegetation index (NDVI) data of the Qinghai–Tibet Plateau.

### 2.3. Methods

#### 2.3.1. Assessment of Ecosystem Services

The core of ecosystem services assessment is the quantification and spatial modeling of ecosystem services. Ecosystem service assessment is mainly based on model assessment, and the commonly used models include Integrated Valuation of Ecosystem Services and Tradeoffs (InVEST) model [34].Artificial Intelligence for Ecosystem Services (ARIES) model [35], Social Values for Ecosystem Services (SolVES) model [36].Their application scope, data requirements and uncertainty of evaluation results are quite different.Fu et al. [37] constructed China’s indicator system for biodiversity and ecosystem service evaluation, and carried out application demonstrations in Inner Mongolia autonomous region, Shenzhen, Lishui, and Pu’er. The current research results establish a direct reference or quantitative relationship between ecosystem service indicators and ecosystem attribute parameters, forming a relatively mature evaluation method, and the evaluation results are highly credible and comparable.

According to the importance of the Qinghai–Tibet Plateau to national and regional ecological security concerns and the availability of data, the three dominant ecosystem functions of water conservation, soil conservation, and windbreak and sand fixation were selected for evaluation [34]. In this study, the water balance equation was used to calculate water conservation [38], the revised universal soil-loss equation (RUSLE) was employedfor soil conservation determination [39], and the revised wind-erosion equation (RWEQ) was used for windbreak and sand fixation characterization [40].

#### 2.3.2. Spatial Differentiation Pattern

The Sen+Mann–Kendall trend test method [41] was used to analyze the change trend and spatial pattern of the ecosystem services of the natural protected areas on the Qinghai–Tibet Plateau in 2000, 2005, 2010, 2015, and 2020 (Table 1). The *Z* scoreis the result of the Mann–Kendall significance test, with |*Z*| ≥ 1.96 indicatingsignificant change and |*Z*| < 1.96 indicatingno significant change. This method providescertain advantages, such as a notableability to avoid errors and the sample not needing to conformto a specific distribution.

#### 2.3.3. Spatial Agglomeration Characteristics

To better understand the spatial agglomeration characteristics of the protection effectiveness of the natural protected areas on the Qinghai–Tibet Plateau, exploratory spatial data analysis (ESDA) was adopted to calculate the global spatial autocorrelation indicator [42], and a map of the local indicators of spatial autocorrelation (LISA) was generated [43]. The global spatial autocorrelation can reveal the spatial agglomeration level of the entire evaluation unit, which is represented by the Moran index and the value ranges from −1 to 1. Local spatial autocorrelation can reflect the local spatial agglomeration characteristics of the evaluation units.

It is calculated as follows:$$I=\frac{w_{ij}\sum_{i=1}^{n}\left(y_{i}-\bar{y}\right)\sum_{j=1}^{n}\left(y_{j}-\bar{y}\right)}{S^{2}\sum_{i=1}^{n}\sum_{j=1}^{n}w_{ij}}$$
where *I* is the Moran index of the evaluation unit; *y_i_* is the observation value in area *i*; *w_ij_* is the proximity relationship between areas *i* and *j*; *S*^2^ is the variance in the observation value of all evaluation units; and $\bar{y}$ is the average of the observations of all evaluation units.

#### 2.3.4. Geodetector Model

In this study, the causal relationship of protection intensity–ecosystem services–main threats was considered, and a system of factors impacting the protection effectiveness of natural protected areas was constructed based on the principles of typicality, quantification, and availability of the influencing factors.

The factors influencingthe protection effectiveness of the natural protected areas on the Qinghai–Tibet Plateau involve categorical variables (such as the grade of natural protected areas and management organization settings). Since the linear regression method is limited to discontinuous variables and is unsuitable for solving theseproblems, the Geodetector model [44] was used to study the driving mechanism of the influencing factors.
$$q=1-\frac{1}{N\sigma^{2}}\sum_{h=1}^{L}N_{h}\sigma_{h}^{2}$$
where *q* is the detection value of the detection factor *x*; *h* (*h* = 1, …, *L*) is the stratification degree of factor *x*; *N* and *N_h_* are the sample numbers of the whole area and the detection area, respectively; and *σ^2^* and *σ_h_^2^* are the *y*-value variances in the whole area and the detection area, respectively. The value range of *q* is [0, 1]. The larger the value of *q* is, the greaterthe effect of *x* on *y*.

The methodological framework is shown in Figure 2.

## 3. Results

### 3.1. Evolutionary Characteristics of the Ecosystem Service Patterns

In 2000, 2005, 2010, 2015, and 2020, the amount of water conservation per unit area of the natural protected area ecosystem on the Qinghai–Tibet Plateau was 336.98 mm, 378.31 mm, 289.55 mm, 300.81 mm, and 285.63 mm, respectively, indicatinga stabletrend after fluctuation and decline (Figure 3a–e). The amount of soil conservation per unit area was 2871.64 t/hm^2^, 3156.74 t/hm^2^, 2837.69 t/hm^2^, 2981.38 t/hm^2^, and 3738.54 t/hm^2^, respectively, indicatinga fluctuating upward trend (Figure 3f–j). The amount of wind prevention and sand fixation per unit area was 36.66 t/hm^2^, 35.80 t/hm^2^, 35.06 t/hm^2^, 32.09 t/hm^2^, and 33.14 t/hm^2^, respectively, indicatingan increasing trend after fluctuation decreases (Figure 3k–o).

### 3.2. Evolution of the Spatial Differentiation Pattern

There were 53 natural protected areas with significant increases in the water conservation function, mainly in Qinghai and Sichuan Provinces. There were 10 natural protected areas with significant declines, mainly in the Tibet Autonomous Region and Yunnan Province. There were 48, 140, and 79 cases with slight decreases, no changes, and slight increases, respectively, accounting for 14.55%, 42.42%, and 23.94%, respectively, of the total number of natural protected areas (Figure 4a). The water conservation function of nature reserves, forest parks, wetland parks, and geoparks remainedunchanged. The water conservation function of desert parks significantly increased, accounting for 58.33% of the total number of desert parks (Figure 4b).

There were 72 natural protected areas with a significantly increased soil conservation function, mainly in Gansu and Sichuan Provinces. There were nine natural protected areas with significant declines, mainly in the Tibet Autonomous Region. There were 27, 140, and 82 cases with slight decreases, no changes, and slight increases, respectively, accounting for 8.18%, 42.42%, and 24.85%, respectively, of the total number of natural protected areas (Figure 4c). The soil conservation function of nature reserves and wetland parks remained unchanged. The soil conservation function of geoparks was slightly increased, the soil conservation function of forest parks remainedeither unchanged or was significantly increased, and that of desert parks remained unchanged, accounting for 91.67% of the total number of desert parks (Figure 4d).

There were 28 natural protected areas with a significantly increased windbreak and sand fixation function, mainly in the Tibet Autonomous Region. There were 23, 159, and 120 cases with slight decreases, no changes, and slight increases, respectively, accounting for 6.97%, 48.18%, and 36.36%, respectively, of the total number of protected areas. No significant reduction occurred (Figure 4e). The windbreak and sand fixation function of nature reserves, forest parks, and desert parks remained unchanged. Wetland parks were dominated by slight increases, and that of geoparks remained largely unchanged or slightly increased (Figure 4f).

### 3.3. Spatial Agglomeration Characteristics

The Moran index valuesof the water conservation, soil conservation, windbreak, and sand fixation functions were all positive and passed the significance test at the 95% confidence level, indicating that the changes in the ecosystem services of the natural protected areas on the Qinghai–Tibet Plateau exhibitedan obvious positive correlation. The spatial agglomeration type was dominated by significant high–high (HH) and low–low (LL)areas, indicating that the phenomenaof high-value agglomeration and low-value agglomeration were prominent (Figure 5). The high-value agglomeration areas of water conservation function change were distributed in the central and eastern Qinghai–Tibet Plateau areas, and the low-value agglomeration areas were distributed in the Tibetan Autonomous Region, Garze Tibetan Autonomous Prefecture of Sichuan, and Three Parallel Rivers in Yunnan. The high-value agglomeration areas of soil conservation function change were distributed in the Aba Tibetan and Qiang Autonomous Prefecture of Sichuan and the Gannan Tibetan Autonomous Prefecture of Gansu, while the low-value agglomeration areas were distributed in the Qilian Mountains, Qinghai Lake, Qaidam area, northern Qinghai–Tibet Plateau, eastern Tibetan alpine canyon area and Three Parallel Rivers. The high-value agglomeration areas of windbreak and sand fixation function change were distributed in the Himalayas, Qilian Mountains, Qinghai Lake, and western Sichuan Plateau, while the low-value agglomeration areas were distributed in the Hengduan Mountains and southern Tibetan valley.

### 3.4. Protection Effectiveness Integration

The spatial differentiation patterns of the water conservation, soil retention, windbreak, and sand fixation functions were superimposed with equal weights. With the use ofthe “natural breaks” method in ArcGIS, the superimposed results were divided into five grades, namely, significantly improved, improved, stable, worse, and significantly worse. Notably, 25 natural protected areas on the Qinghai–Tibet Plateau were graded as attaining a significantly improved protection effectiveness, 151 natural protected areas were graded as improved, 84 natural protected areas were graded as stable, 56 natural protected areas were graded as worse, and 14 natural protected areas were graded as significantly worse, accounting for 1.55%, 10.05%, 21.41%, 40.32%, and 26.67%, respectively, of the natural protected areas (Figure 6). The areas with significantly improved protection effectiveness were mainly located in Gansu and Sichuan, such as Gansu Bailongjiang Axia Provincial Nature Reserve, Sichuan Gonggangling Provincial Nature Reserve, and Sichuan Sanao Snow Mountain Forest Park. The areas with significantly worse protection effectiveness were concentrated in Tibet and Yunnan, such as the Tibet Mangkang Yunnan Snub-Nosed Monkey National Nature Reserve, Yunnan Baima Snow Mountain National Nature Reserve, and Yunnan Gaoligong Mountain National Nature Reserve.

### 3.5. Identifying the Factors Influencing the Protection Effectiveness

(1) Establishment of the indicator system of influencing factors

According to the natural geography, protection intensity, and development and construction status of the natural protected areas on the Qinghai–Tibet Plateau, 11 influencing factors, namely precipitation change (PC), temperature change (TC), altitude (AL), grade of the natural protected area (GPA), management organization setting (MOS), approval time (AT), population density (PD), built-up area density (BAD), commercial forest density (CFD), cultivated land density (CLD), and mining area density (MAD), were screened to establish an indicator system for protection effectiveness. Withinthe context of climate change, temperature rise, precipitation change, and plant community succession are the basic driving forces of ecosystem service changes in natural protected areas. In this study, precipitation and temperature were selected as the main driving factors of climate change. The altitude of the Qinghai–Tibet Plateau variesbetween 3000 and 5000 m, and the vertical zonality of the geomorphology is obvious. Altitude is the main influencinggeographical factor. The protection intensity factor is an important positive driving factor of the protection effectiveness of natural protected areas. Generally, the longer the construction and management time and the higher the grade of the protected area, the more obvious the protection effectiveness is. In thisstudy, the grade of the protected area, the setting of management agencies, and approval time were introduced as covariates to verify the impact of the protection intensity on the protection effectiveness of natural protected areas. To quantify the impact of development and construction on natural protected areas, the population density, built-up area density, commercial forest density, cultivated land density, and mining area density were introduced as reverse influencing factors.

The input variables of the Geodetector modelcomprise categorical data, and continuous variables must be discretized [45]. The grades of protected areas were divided into four grades according to county, city, provincial, and national levels; the setting of management agencies was divided into two grades according to whether they were established or not. With the use of the Sen+Mann–Kendall trend test method, the changes in water conservation, soil conservation, windbreak and sand fixation, precipitation, and temperature were graded. With the use of the “natural breaks” method in ArcGIS and the expert scoring method, the other influencing factors were graded (Table 2, Figure 7).

(2) Identification of the influencingfactors

The Fishnet tool in ArcGIS was used to extract and transformraster data into points, and the sampling interval was 5 km. A total of 32,405 points were extracted as the operation data of the Geodetector model. The factor detection results showed that the explanatory power of the factors influencing the protection effectiveness of the natural protected areas on the Qinghai–Tibet Plateau in descending order was precipitation change > altitude > mining area density > temperature change > cultivated land density > construction approval time > population density > grade of the natural protected area > commercial forest density > built-up area density > management agency setting (Table 3). The interactive detection results showed that the effect of the interaction betweenany two variables on the protection effectiveness of natural protected areas was greater than the independent effect. The effects of precipitation change and altitude on the protection effectiveness of natural protected areas significantly differedfrom those of the other influencing factors (Table 3). The protection effectiveness index valuesof the natural protected areas at altitudes below 3500 m, 3500–4000 m, 4000–4500 m, 4500–5000 m, and above 5000 m were 4.29, 4.12, 3.96, 3.73, and 3.64, respectively. This result indicated that with increasing altitude, the protection effectiveness of natural protected areas gradually decreased.

The risk detector results (Table 4) showed that the protection effectiveness of earlier established natural protected areas (before 2000), national natural protected areas, and established management agenciesand the effects of the lower built-up area density (<0.75%) and lower mining area density (<1%) were remarkable.

## 4. Conclusions and Discussion

Physical and geographical factors are the basic driving forces that affect the protection effectiveness of natural protected areas. The protection intensity exertsa significant impact on the protection effectiveness of natural protected areas. In terms of the protection level, there are 152 national-level natural protected areas, accounting for 46.06% of the total number of natural protected areas and 83.01% of the total area of natural protected areas on the Qinghai–Tibet Plateau. There are 178 local-level natural protected areas, accounting for 53.94% (number) and 16.99% (area), respectively. In terms of management institutions, 221 natural protected areas are associated with established management institutions, accounting for 66.97% of the total number of natural protected areas. The naturalprotected areas in the Tibet Autonomous Region and Xinjiang and Gansu Provinces are associated withestablished management institutions, and most desert parks, forest parks, and wetland parks in Qinghai are not associated with established management institutions. Provincial-level wetland parks and county-level nature reserve management institutions are seriously insufficient in Sichuan Province. In terms of the approval time, 292 natural protected areas were established before 2000. In addition, individual development and construction factors, such as the density of mining and built-up areas, significantly impact the effectiveness of natural protected areas. The naturalprotected areas in the Tibet Autonomous Region and Qinghai Province encompassthe largest mining areas, and the natural protected areas in Sichuan Province attainthe highest mining density. The natural protected areas in the Tibet Autonomous Region and Qinghai Province havethe largest built-up areas, and the natural protected areas in Sichuan and Yunnan Provinces exhibitthe highest density of built-up areas. In conclusion, the influencing factors and characteristics of protection effectiveness identified in this study could provide support for promoting the scientific establishment of various types of natural protected areas at all levels.

There are uncertainties in the evaluation methodology adopted in this study.The input variables of the Geodetector model must comprise categorical data, and continuous variables must be discretized. The classification effect can be evaluated using the detection factor statistical q value of the geographic detector operation results, and the larger the q value is, the better the classification assignment of the influencing factors is. After employing classification algorithms such as the K-means algorithm, the equidistant method, and the quantile method, it was decided to apply the “natural breaks” clustering method and expert scoring method to assign grades to the influencing factors such as altitude, approval time, and population density. In the future, we should strengthen the analysis of discretization processing methods to further improve the reliability of the Geodetector model. Basedon a large number of field observations and experimental analyses, the parameters related to the evaluation of ecosystem services should be localized in the study area, and the rationality of the parameters should be verified to further enhance the accuracy of the evaluation of ecosystem services on the Qinghai–Tibet Plateau.

At present, most assessments based on ecosystem services are mainly retrospective assessments. In the future, the temporal and spatial evolution features of ecosystem services can be simulated and predicted under scenarios of different driving factors and management and control measures to achieve the precise management and control of the influencing factors and improve the effectiveness of ecological risk management. Comparative analysis of the protection effectiveness of the ecological environment and biodiversity inside and outside natural protected areas can be evaluated, and the impact and driving mechanism of natural protected areas on the surrounding area can be examined. The Qinghai–Tibet Plateau is one of the regions with the richest biodiversity in the world, with more than 3760 species of endemic seed plants and more than 280 species of vertebrates, including more than 300 species of rare and endangered higher plants and more than 120 species of rare and endangered animals [46]. In the study area, it is urgent to carry out regional biodiversity surveys and assessments to elucidatethe distribution, change, and threatened status of biodiversity and to incorporate biodiversity into the assessment of the protection effectiveness of natural protected areas. We should explore and study the technical specifications for the evaluation of the protection effectiveness of natural protected areas, promote the realization ofthe standardized and efficient construction management of natural protected areas, and performthe normalized evaluation of the protection effectiveness of natural protected areas.

## Figures and Tables

**Figure 1 ijerph-20-02605-f001:**
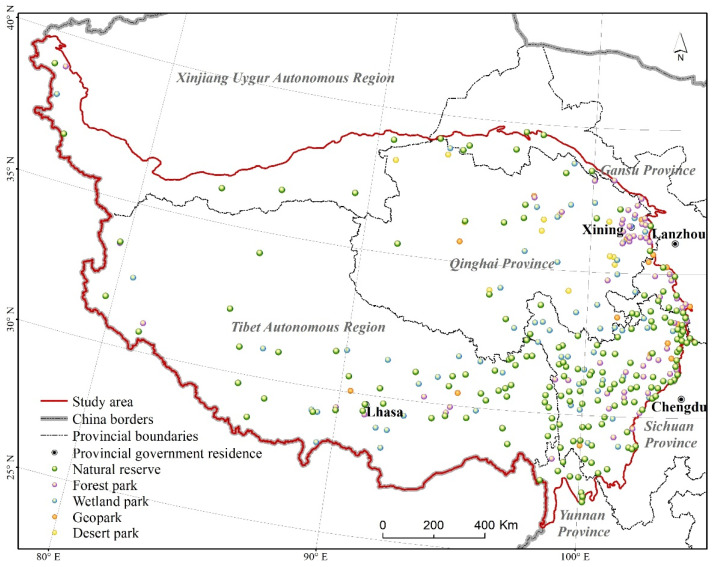
Spatial distribution of the natural protected areas on the Qinghai–Tibet Plateau.

**Figure 2 ijerph-20-02605-f002:**
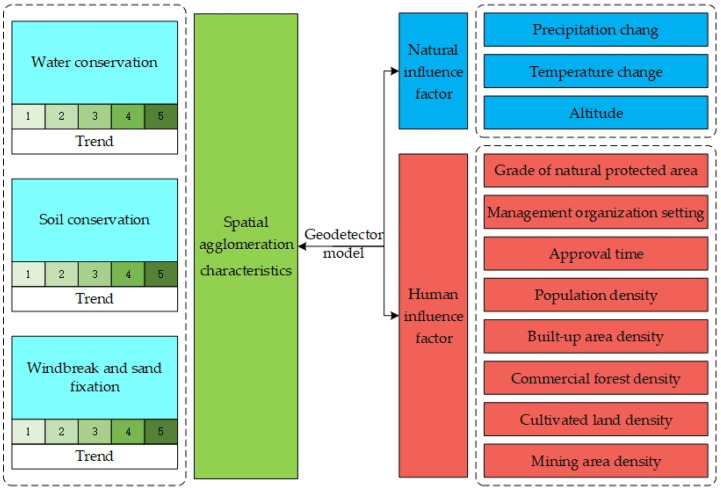
Methodological framework.

**Figure 3 ijerph-20-02605-f003:**
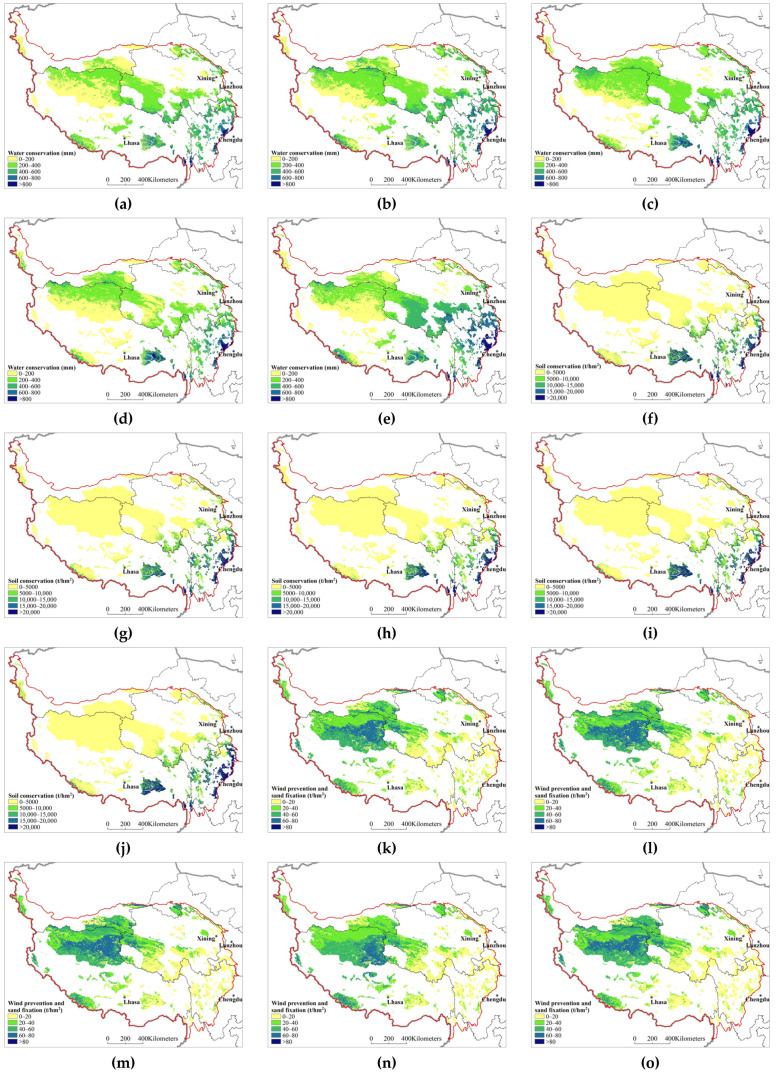
Temporal and spatial changes in the ecosystem services of the natural protected areas on the Qinghai–Tibet Plateau; (**a**)—water conservation in 2000; (**b**)—water conservation in 2005; (**c**)—water conservation in 2010; (**d**)—water conservation in 2015; (**e**)—water conservation in 2020; (**f**)—soil conservationin 2000; (**g**)—soil conservation in 2005; (**h**)—soil conservation in 2010; (**i**)—soil conservation in 2015; (**j**)—soil conservation in 2020; (**k**)—wind prevention and sand fixationin 2000; (**l**)—wind prevention and sand fixation in 2005; (**m**)—wind prevention and sand fixation in 2010; (**n**)—wind prevention and sand fixation in 2015; (**o**)—wind prevention and sand fixation in 2020.

**Figure 4 ijerph-20-02605-f004:**
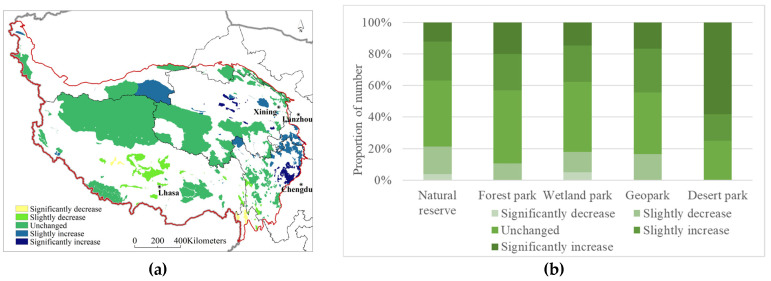
Evolution of the spatial pattern of ecosystem services of the natural protected areas on the Qinghai–Tibet Plateau; (**a**)—spatial pattern of water conservation; (**b**)—trend statistics of water conservation; (**c**)—spatial pattern of soil conservation; (**d**)—trend statistics of soil conservation; (**e**)—spatial pattern of wind prevention and sand fixation; (**f**)—trend statistics of wind prevention and sand fixation.

**Figure 5 ijerph-20-02605-f005:**
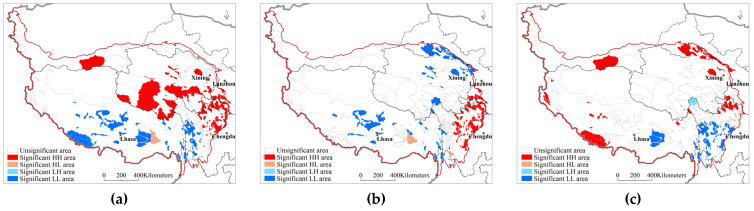
Spatial clustering features of the protection effectiveness of the natural protected areas on the Qinghai–Tibet Plateau based on ecosystem services. (**a**) Water conservation (**b**) Soil conservation (**c**) Windbreak and sand fixation.

**Figure 6 ijerph-20-02605-f006:**
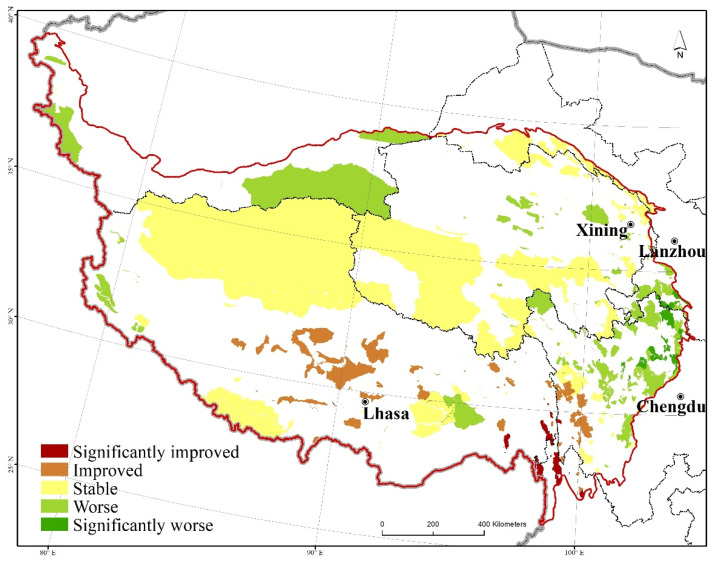
Spatial distribution of protection effectiveness of the protected areas of the Qinghai–Tibet Plateau.

**Figure 7 ijerph-20-02605-f007:**
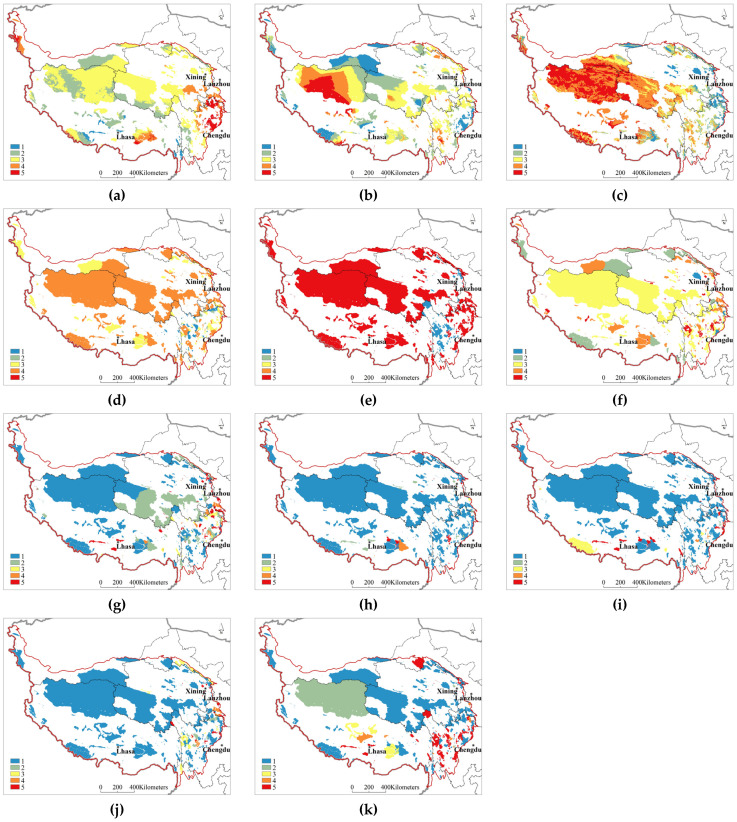
Spatial distribution of the influencing factors of the protection effectiveness of the natural protected areas on the Qinghai–Tibet Plateau. (**a**) Precipitation (**b**) Temperature (**c**) Altitude (**d**) Grade of natural protected area (**e**) Management organization (**f**) Approval time (**g**) Population density (**h**) Built-up area density (**i**) Commercial forest (**j**) Cultivated land density (**k**) Mining area density.

**Table 1 ijerph-20-02605-t001:** Classification of the changes in the ecosystem services of the protected areas on the Qinghai–Tibet Plateau.

Sen	Mann–Kendall	Trend
<0	|Z| > 1.96	Significantly reduced
<0	|Z| ≤ 1.96	Slightly reduced
=0		Unchanged
>0		Slightly increased
>0	|Z| > 1.96	Significantly increased

**Table 2 ijerph-20-02605-t002:** Classification of the influencing factors of the protection effectiveness of the natural protected areas on the Qinghai–Tibet Plateau.

Influencing Factors	1	2	3	4	5
Water conservation change	Significantly decrease	Slightly decrease	Unchanged	Slightly increase	Significantly increase
Soil conservation change	Significantly decrease	Slightly decrease	Unchanged	Slightly increase	Significantly increase
Windbreak and sandfixation change	Significantly decrease	Slightly decrease	Unchanged	Slightly increase	Significantly increase
PC	Significantly decrease	Slightly decrease	Unchanged	Slightly increase	Significantly increase
TC	Significantly decrease	Slightly decrease	Unchanged	Slightly increase	Significantly increase
AL (m)	<3500	3500–4000	4000–4500	4500–5000	>5000
AT (year)	<1980	1980–1990	1991–2000	2001–2010	2011–2020
PD(people/km^2^)	0–2	2–4	4–6	6–8	>8
BAD (%)	0–0.25	0.25–0.5	0.5–0.75	0.75–10	1–100
CFD (%)	0–0.15	0.15–0.25	0.25–0.35	0.35–0.45	0.45–100
CLD (%)	0–0.25	0.25–0.5	0.5–0.75	0.75–1	1–100
MAD (%)	0–0.25	0.25–0.5	0.5–0.75	0.75–1	1–100

**Table 3 ijerph-20-02605-t003:** Interactive detection results of the influencing factors of the protection effectiveness of the protected areas on the Qinghai–Tibet Plateau.

	GPA	MAD	AL	CLD	MOS	BAD	PC	AT	PD	CFD	TC
*q*	0.010	0.037	0.040	0.024	0.001	0.006	0.177	0.018	0.012	0.009	0.029
GPA	0.010										
MAD	0.067	0.037									
AL	0.053	0.089	0.040								
CLD	0.036	0.067	0.063	0.024							
MOS	0.015	0.042	0.045	0.028	0.001						
BAD	0.020	0.051	0.053	0.035	0.008	0.006					
PC	0.186	0.246	0.204	0.187	0.190	0.181	0.177				
AT	0.051	0.080	0.064	0.051	0.020	0.029	0.196	0.018			
PD	0.041	0.062	0.065	0.041	0.018	0.025	0.203	0.042	0.012		
CFD	0.034	0.057	0.060	0.042	0.015	0.021	0.192	0.039	0.035	0.009	
TC	0.055	0.095	0.081	0.055	0.040	0.039	0.209	0.055	0.047	0.049	0.029

**Table 4 ijerph-20-02605-t004:** Risk detection results of the factors influencing the protection effectiveness of the protected areas on the Qinghai–Tibet Plateau.

MOS		0	1			
	0					
	1	Y *^a^*				
GPA		1	2	3	4	
	1					
	2	Y				
	3	N *^b^*	Y			
	4	Y	Y	Y		
AT		1	2	3	4	5
	1					
	2	Y				
	3	Y	Y			
	4	Y	Y	Y		
	5	Y	N	Y	Y	
BAD		1	2	3	4	5
	1					
	2	Y				
	3	Y	Y			
	4	Y	Y	N		
	5	N	Y	Y	Y	
MAD		1	2	3	4	5
	1					
	2	Y				
	3	Y	Y			
	4	Y	Y	Y		
	5	Y	N	Y	Y	
AL		1	2	3	4	5
	1					
	2	Y				
	3	Y	Y			
	4	Y	Y	Y		
	5	Y	Y	Y	Y	

*^a^* Y indicates a significant difference. *^b^* N indicates no significant difference.

## Data Availability

Not Applicable.

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
