# Peer review of "Evaluation of the Protection Effectiveness of Natural Protected Areas on the Qinghai–Tibet Plateau Based on Ecosystem Services"

_ijerph, 2023, doi:10.3390/ijerph20032605_

Round 1

Reviewer 1 Report

Dear authors,

thanks for the paper, the topic is interesting, but there is a need to make large improvements.  I have some suggestions:

1.       Please extend the introduction with a focus on more deep study of methods for the evaluation of ecosystem services. This topic is broadly discussed in the literature, but the broader Analysis of these sources is missing. Please expand this.

2.       In methodology is not clear the selection of user data, seems, that sources are limited for complex ekosystém services evaluation

3.       The methods is purely described and it is not fully clear the real methodology

4.       Again Ecosystem service spart needs to be extended

5.       Please describe in detail how pattern changes were evaluated

6.       Please explain, how Aglomaration characteristics are related to ekosystém services.

7.       The same for geodetector model

8.       Not clear, how Water conservation per unit is connected with Methodology,

Reviewer 2 Report

An interesting and well constructed study. The literature review is comprehensive and appropriate to the study. The Methodology is well developed and should provide guidance to future researchers. The findings should also be useful in informing policy makers.

A reference that invited a query is "Joppa and Pfaff (2011) evaluated the protection effectiveness of 39 natural protected areas in 147 countries". This is virtually all countries in the world. Is this correct?

Reviewer 3 Report

The topic of this manuscript is timely and potentially interesting for readers of IJERPH. Overall, the manuscript is well-written and structured. But I have some comments, so I decided to give a chance to authors for revising the manuscript. 

General Comments:

1. I found that various and important methodological details were not described in the manuscript.
2. Furthermore, the authors should also have considered in the introduction and the discussion recent literature on "Natural protected areas". Some of the important and recent studies outside of China are (be aware, I am not one of the authors of the following papers): 

** https://doi.org/10.3390/land11101871 (it's about the role of the local community in managing protected areas).

** https://doi.org/10.1016/j.regsus.2022.03.001 (A safeguard strategy in protected areas)

** https://doi.org/10.3390/su141710956 (it's about the spatiotemporal of natural capital in protected areas).

The authors only used 10 papers in the introduction section, and only one of them is an updated reference. 

3. Finally, a section about the limitation of the methodological approach should be added before the conclusions.

Minor comments:

L 49-50: "While most of the studies focused on the protection effectiveness evaluation of nature reserves (Add some citations here)"

L 59-60: Need a citation.

L 104:-106: Add some citations about the application of the Fao Penman method in various countries (for example https://doi.org/10.17221/83/2016-JFS, Iran as a case study; or this study: https://doi.org/10.3390/hydrology9110206 for Siri Lanka).

L 170: "Where", change it to "where"

At the end of Materials and Methods, I would like to see a flowchart to better understand the author's approach.

L 327-369: The discussion part is soo poor. There is no comparison with past research, even no interpretation fo results. Just representing results as a discussion is not acceptable. Also, I encourage authors to create a conclusion section

Round 2

Reviewer 1 Report

Dear authors,

thanks for updated version, there is important progress, but till now I see as important to go to more deep literature review. Ecosystem services evaluation are often described in publications and better overview is missing. Look please more on studies in other areas, not only for your region.
